# The effect of carbon fertilization on naturally regenerated and planted US forests

Eric C. Davis [1] ✉, Brent Sohngen [2] & David J. Lewis [3]

Over the last half century in the United States, the per-hectare volume of wood in trees has increased, but it is not clear whether this increase has been driven by forest management, forest recovery from past land uses, such as agriculture, or other environmental factors such as elevated carbon dioxide, nitrogen deposition, or climate change. This paper uses empirical analysis to estimate the effect of elevated carbon dioxide on aboveground wood volume in temperate forests of the United States. To accomplish this, we employ matching techniques that allow us to disentangle the effects of elevated carbon dioxide from other environmental factors affecting wood volume and to estimate the effects separately for planted and natural stands. We show that elevated carbon dioxide has had a strong and consistently positive effect on wood volume while other environmental factors yielded a mix of both positive and negative effects. This study, by enabling a better understanding of how elevated carbon dioxide and other anthropogenic factors are influencing forest stocks, can help policymakers and other stakeholders better account for the role of forests in Nationally Determined Contributions and global mitigation pathways to achieve a 1.5 degree Celsius target.

In the United States, over the last two decades, a large forest C-sink has sequestered 700–800 million tons of $CO_2$ per year, which is roughly 10–11% of US gross $CO_2$ emissions[1–3]. In addition, the per-hectare volume of trees has increased over the last 50 years[4]. Among the ten forest groups in this study, all except Aspen-Birch have increased their per-hectare wood volume from 1997 to 2017 (Fig. 1)[5,6].

It is not clear, however, whether the tree-volume and C-sink increase have been driven by forest recovery from past land uses, such as agriculture[7], or other environmental factors such as elevated $CO_2$, N deposition, or climate change[8]. Even less well understood is the role of forest management through planting, harvesting of secondary and old-growth forests, and forest management that is certified with sustainability criteria, even though these approaches are employed on an increasingly large share of the world's forests[9]. As policymakers look to the future, where forests are expected to play a large role in Nationally Determined Contributions (NDCs) and global mitigation pathways to achieve a 1.5 °C target[10–13], it is important to better understand how

anthropogenic factors like elevated $CO_2$ may influence current forest stocks and the outcomes of forest-expansion (e.g., afforestation) and conservation policies (e.g., avoided deforestation).

This analysis builds on the experimental and modeling literature that has examined the role of elevated $CO_2$, like the FACE studies that controlled $CO_2$ levels on plots within the same location and found that elevated $CO_2$ increased net primary productivity (NPP)[14,15]. The effects these studies observed, however, might not scale up across ecosystems and over time with disturbances and other processes at play.

Tree ring studies[16,17] offer the potential to observe the effects of elevated $CO_2$, but with limited observations and controls, they could not identify the effects of elevated $CO_2$ separately from other factors, such as weather. Other research has shown that tree heights in Poland increased over time, although the role of $CO_2$ concentration was not identified[18], and that $CO_2$ exposure generates smaller effects on tree-volume in settings that are colder and have more water stress[19]. A recent meta-analysis of experimental results estimated that each

[1]United States Department of Agriculture-Economic Research Service, Kansas City, MO 64105, USA. [2]Department of Agricultural, Environmental, and Development Economics, The Ohio State University, Columbus, OH 43210, USA. [3]Department of Applied Economics, College of Agricultural Sciences, Oregon State University, Corvallis, OR 97331, USA. ✉e-mail: eric.davis3@usda.gov

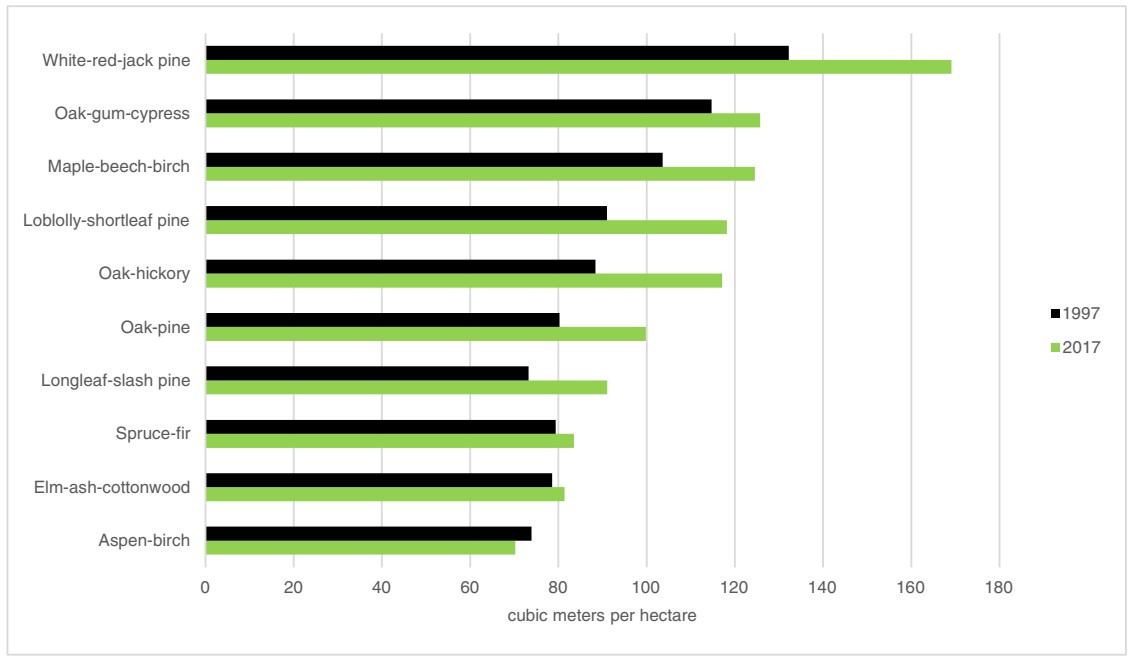

**Fig. 1 | Wood volume per hectare in 1997 and 2017 separated by forest group.** Source: USFS 1997 and 2017 RPAs[5,6].

100-ppm increase in $CO_2$ increases aboveground volume in ecosystems by 8.2%[20]. These experimental results underpin many dynamic global vegetation models (DGVMs), but they have not been replicated with observations from plots drawn randomly within ecosystems. This study addresses that gap by using quasi-experimental techniques[21] and data drawn from tens of thousands of plots in the United States to identify, specifically and differentially, the role of elevated $CO_2$ from other factors that also influence wood volume in forest stands.

We show that there has been a positive response in wood volume to elevated $CO_2$ in ten temperate forest groups in the United States. Moreover, our estimated response in ecosystems is larger than that predicted by experimental results. Those estimates, however, included non-forested ecosystems, which do not appear to respond as strongly as forests to elevated $CO_2$. For the forest groups in our study with both planted and naturally regenerated observations, we find that the proportional impact of elevated $CO_2$ on planted stands is roughly equivalent to the impact in natural stands, but the results in cubic meters per hectare are mostly larger for planted stands because planted stands, on average, have more volume than naturally regenerated stands.

## Results

This study estimates the impact of elevated $CO_2$ on wood volume in the central stems of trees by comparing growing-stock volume on recently observed timber plots with matched plots observed decades prior using historical data from the US Forest Service (USFS) Forest Inventory and Analysis Program (FIA)[22]. A key challenge in understanding how elevated $CO_2$ affects the wood volume observed in forest plots is that $CO_2$ concentrations change only temporally but not spatially. That means that all forests are exposed to the same $CO_2$ concentration each year. Our research design overcomes this challenge by utilizing the fact that forests embed the historical $CO_2$ concentrations to which they have been exposed. Thus, two plots of the same age and type of forest observed at two different time periods have effectively received two different $CO_2$-exposure profiles. This is also true when two plots of the same type of forest are observed at the same time period, but their ages differ.

Since other factors such as climate, technology, pests, and tree stocking also affect the volume and vary over time, we use modern econometric techniques that combine matching with post-matching regression analysis, which includes time-fixed effects, to isolate the effects of elevated $CO_2$ from other time-varying and time-invariant covariates that might also affect tree volumes[23]. These matching methods[24] systematically construct treatment and control groups of timber plots that are similar in observable characteristics, but that vary in their exposure to $CO_2$ due to the observations having been taken in different decades.

Forests provide several advantages to facilitate empirical identification of the impact of elevated $CO_2$ on tree wood volume. First, volume and management decisions on US forests have been systematically monitored and measured for many decades through the plot-level FIA database. Second, the FIA data indicate whether forest plots were regenerated naturally or through active planting. While the wood volume of several species of commercially planted conifers have been affected by changing seed technology over the last few decades, naturally regenerated forests—including most hardwoods—have not been influenced by such technological changes that would confound the identification of the impact of elevated $CO_2$.

We examine ten temperate forest-type groups in the United States whose range in the FIA database is shown in Fig. 2. First, we focus only on observations of naturally regenerated (unmanaged) forest plots because natural stands arguably have not been affected by the advances in tree planting and seed selection that have improved the quality of planted stands over the same time period. To address the potential for bias in our data, we follow recent advances in quasi-experimental econometrics and "trim" the data using matching methods[23] to create a dataset of one-to-one matches between a control group comprised of observations taken by the USFS during the pre-1990 (low $CO_2$) period and a treatment group whose observations were made in the post-2000 (high $CO_2$) period, while controlling for multiple salient covariates that are correlated with time and/or could affect yield, such as age and stocking conditions (Supplementary Tables 1–6 and Supplementary Data 1, 2). A variety of alternative functional forms are tested (Supplementary Data 3–10), with results robust across specifications. The effect of elevated $CO_2$ is identified by the natural log of lifetime $CO_2$, which is the logged sum of the annual $CO_2$ concentrations experienced by each stand-up to its age class at the time the plot is measured. Climate-related effects on wood volume are captured with seasonal temperature and precipitation variables.

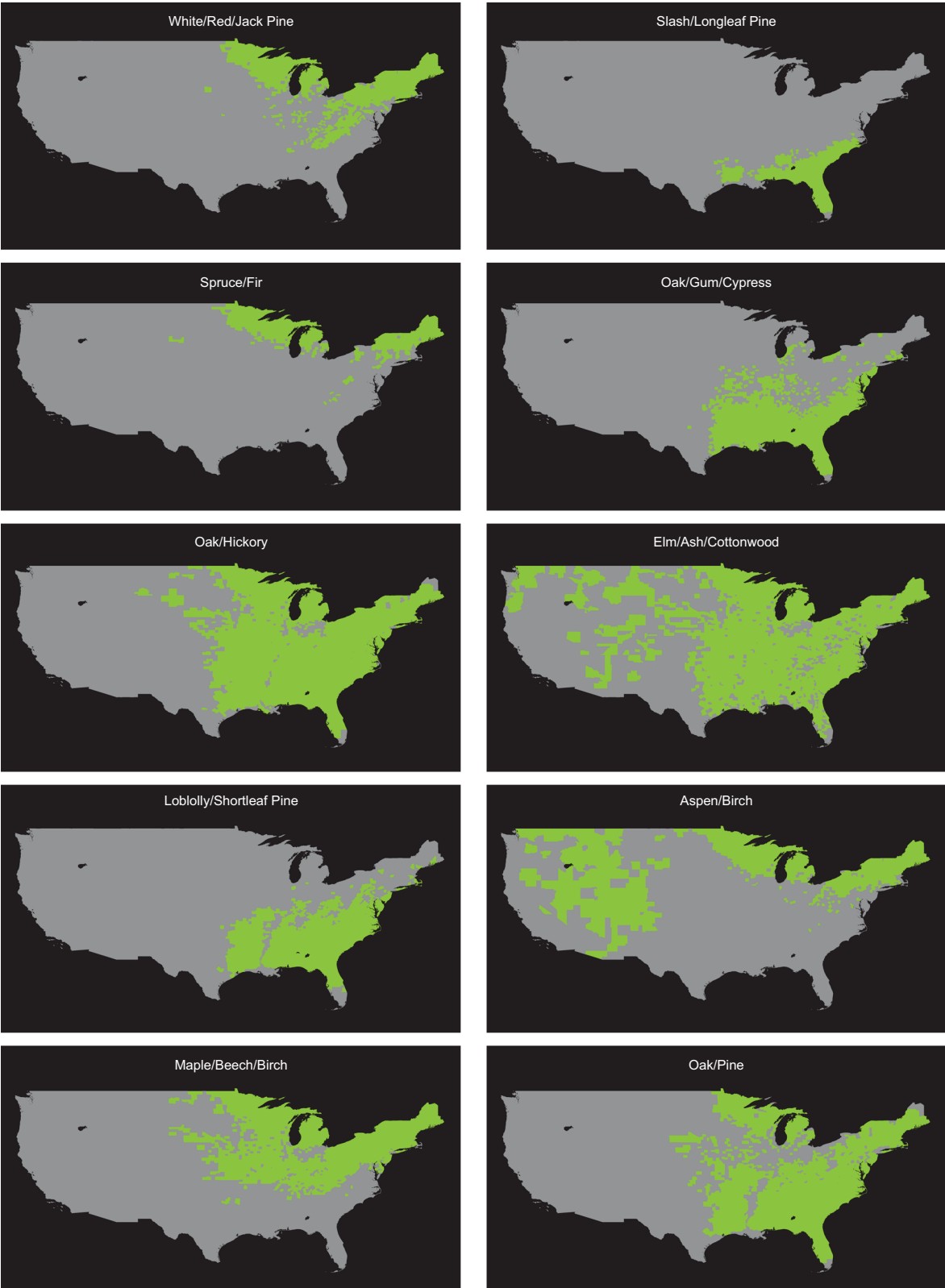

**Fig. 2 | Geographic range of forest groups based on observations taken by USFS.** Note: This figure for each forest group details (in green) all counties in which the US Forest Service Forest Inventory and Analysis (USFS-FIA) database has recorded the forest group's presence between 1968 and 2018. This is based on its annual resource inventories and is limited to observations of stands between 1 and 100 years of age.

To strengthen the identification of elevated $CO_2$, other environmental factors correlated with time, such as nitrogen deposition, large-scale disturbances associated with invasive species (e.g., Emerald Ash Borer, *Agrilus planipennis Fairmaire*), and widespread phenomena like Sudden Aspen Death, are captured by including time-dummy variables (fixed effects). For Table 1, the dummy variable is designed to capture changes between the pre-1990 and the post-2000 periods that are not tied to the gradually increasing $CO_2$ trendline (e.g., the impact of nitrogen deposition). The approach is detailed in Model (1) of Supplementary Data 11 and the multivariate-regression results are presented as Model (1) in Supplementary Data 12–22. In six of the ten forest groups, two-sided *t*-tests indicate that the time-dummy coefficients are significant with two forest groups positively impacted and four negatively (Table 1). Aspen/Birch appears to have experienced the largest decline over time (−18.0%; $P < 0.01$; 99% CI from −30.8 to −5.2%; $t = −3.62$; $df = 16{,}416$), a result potentially due to aspen decline[25–28]. Elm/Ash/Cottonwood also decreases (−11.8%; $P < 0.01$; 99% CI from −23.1 to −0.6%; $t = −2.7$; $df = 7562$), a result that might be the result of the Emerald Ash Borer that has plagued Ash trees in the Eastern US from 2002 to the present[29,30]. White/Red/Jack pine shows the largest increase (21.2%; $P < 0.01$; 99% CI from 0.1 to 42.2%; $t = 2.6$; $df = 12{,}458$) while Spruce/Fir experienced a 19.2% increase ($P < 0.10$; 90% CI from 1.1 to 37.2%; $t = 1.8$; $df = 4926$), a result most likely due to recovery from spruce budworm (*Choristoneura fumiferana*) outbreaks in the 1970s and early 1980s[4,31]. Overall, across all forests, these large-scale changes from episodic phenomena have reduced wood volume by an average of 8.9% ($P < 0.01$; 99% CI from −10.7 to −7.0; $t = −12.4$; $df = 123{,}145$).

As for elevated $CO_2$, results indicate that for each of the ten forest groups, it has had a positive and significant effect ($P < 0.01$) on growing-stock volume. In Supplementary Data 12–21: Models (1–4), the parameters for logged cumulative lifetime $CO_2$ exposure range from 0.92 to 1.42, with an average across all species and models of 1.15. The magnitude of this $CO_2$ exposure parameter indicates that each 1% increase in lifetime $CO_2$ exposure, which is an additional 3.2–3.3 ppm $CO_2$[32–34] or 2.2–3.4 years' worth of net global emissions over the period from 1984 to 2010 (the midpoints of our control and treatment groups), leads to a 1.15% increase in wood volume.

When examining the cumulative impact of elevated $CO_2$ and analyzing the impact by forest group by age, we estimate that between 1970 and 2015, there has been a significant increase in the wood volume of trees (Table 1). For example, at 75 years of age, the magnitude of the increase is smallest for Maple/Beech/Birch at 9.9% (+15.6 m³ ha⁻¹; $P < 0.01$; 99% CI from 5.3 to 14.6%; $t = 5.0$; $df = 12{,}458$) and largest for White/Red/Jack pine at 14.4% (+26.3 m³ ha⁻¹; $P < 0.01$; 99% CI from 0.4 to 28.3%; $t = 2.4$; $df = 2307$). Overall, the change in volume is significant at $P < 0.01$ across all ages and for all forest groups, except Spruce/Fir. For Spruce/Fir, the estimated change is significant ($P < 0.01$; $df = 4926$) for 25-year-old stands at 17.4% (+5.8 m³ ha⁻¹; 99% CI from 5.0 to 29.9%; $t = 3.3$) and for 50-year-old stands at 13.2% (+9.0 m³ ha⁻¹; 99% CI from 0.8 to 25.7%; $t = 2.5$) but slightly less significant ($P < 0.05$) for 75-year-old stands at 10.6% (+10.0 m³ ha⁻¹; 95% CI from 1.8 to 19.4%; $t = 2.0$). This likely reflects the fact that the proportional increase in volume due to elevated $CO_2$ is greatest for younger stands in Table 1 because younger stands experienced a larger proportional change in $CO_2$ exposure from 1970 to 2015. For instance, between 1970 and 2015, cumulative lifetime $CO_2$ increased 11.1% for 75-year-old stands, whereas for 25-year-old stands, it increased 18.9%[32–34].

Next, we combine all the matched data for naturally regenerated stands, add forest-type controls to account for the heterogeneity across the forest groups, and estimate the exponential volume function, an approach shown as Model (2) of Supplementary Data 11. Results (Supplementary Data 22) show that the average impact of elevated $CO_2$ on the volume of 75-year-old trees across all ten forest groups from 1970 to 2015 is +12.3% (+23.0 m³ ha⁻¹; $P < 0.01$; 99% CI from 7.9 to 16.7%; $t = 6.5$) (Table 1).

Given that the Western US faces different threats and disturbances than the Eastern US, we repeat the process used to create Table 1 with only plots from the 33 states the USFS considers part of the East. Specifically, this comprises all states fully east of the 100th meridian plus Texas and Oklahoma. Balance statistics are provided in Supplementary Data 23. Supplementary Data 11 details the models used for each forest group individually (Model 3) and the combined sample (Model 4). Regression results are presented in Supplementary Data 24, 25. To enable ease of comparison with Table 1, a revised version for the Eastern US is also included (Supplementary Table 7). Results for the four forest groups whose original match results had no observations from the Western US, of course, are identical. Among the remaining six, there are slight changes in the impact of elevated $CO_2$. At age 75, Elm/Ash/Cottonwood increases from 10.0% (16.0 m³ ha⁻¹) for all observations to 11.0% (18.0 m³ ha⁻¹; $P < 0.01$; 99% CI from 4.7 to 17.2%; $t = 4.1$; $df = 6724$) for observations in the Eastern US. Oak/Pine increases from 12.6% (25.1 m³ ha⁻¹) to 13.1% (26.7 m³ ha⁻¹; $P < 0.01$; 99% CI from 7.4 to 18.7%; $t = 5.4$; $df = 9814$), and Aspen/Birch increases from 12.1% (17.6 m³ ha⁻¹) to 12.4% (18.1 m³ ha⁻¹; $P < 0.01$; 99% CI from 8.0 to 16.8%; $t = 6.6$; $df = 16{,}349$). Maple/Beech/Birch increases from 9.9% (15.6 m³ ha⁻¹) to 10.0% (15.8 m³ ha⁻¹; $P < 0.01$; 99% CI from 5.4 to 14.7%; $t = 5.0$; $df = 12{,}360$). Spruce/Fir decreases from 10.6% (10.0 m³ ha⁻¹) to 10.5% (9.8 m³ ha⁻¹; $P < 0.05$; 95% CI from 1.9 to 19.0%; $t = 2.0$; $df = 4930$) and Oak/Hickory remains the same at 11.5% (19.6 m³ ha⁻¹; $P < 0.01$; 99% CI from 9.0 to 14.1%; $t = 10.6$; $df = 38{,}726$). As to the coefficients capturing episodic phenomena, they keep the same sign and differences in effect size are modest.

For a final robustness check, we test an alternative volume function and nonlinear least squares regression approach, detailed in Supplementary Data 11 (Models 5–6), that uses the same matched samples as Table 1. Results from these regressions are presented in Supplementary Data 26, 27 and the impacts are similar in sign and magnitude to those shown in Table 1 (Supplementary Tables 8, 9).

We next examine the impact of elevated $CO_2$ separately for naturally regenerated and planted stands for the three forest groups where sufficient data of both types is available (Supplementary Data 11: Model 7). Since many replanted stands are harvested in relatively short timber rotations, meaning there are few stands older than age 50, we limit the data for both to ages less than or equal to 50 years to better compare the different stands[35–40]. Because of this change in age classes, we create new matches for both naturally regenerated and planted stands (Supplementary Data 28–30) and then run new post-matching regressions (Supplementary Data 31–33). Results indicate that elevated $CO_2$ has positive effects ($P < 0.01$) on wood volume for each forest type examined on both natural and planted stands. Based on these regressions, the predicted change in volume due to elevated $CO_2$ at 25 years of age for planted stands in 1970 versus 2015 (Table 2) for Slash/Longleaf pine is +15.5 m³ ha⁻¹ (+21.7%; $P < 0.05$; 95% CI from +1.8 to +29.3 m³ ha⁻¹; $t = 1.9$; $df = 859$). For Loblolly/Shortleaf pine, the change is +34.6 m³ ha⁻¹ (+27.4%; $P < 0.05$; 95% CI from +17.4 to +51.8 m³ ha⁻¹; $t = 3.9$; $df = 2757$), and for White/Red/Jack pine, the change is +16.0 m³ ha⁻¹ (+42.8%; $P < 0.05$; 95% CI from +2.1 to +29.8 m³ ha⁻¹; $t = 2.3$; $df = 671$). For naturally regenerated stands, the gains are smaller. For Slash/Longleaf pine, the change is +12.5 m³ ha⁻¹ (+21.7%; $P < 0.05$; 95% CI from +4.3 to +20.7 m³ ha⁻¹; $t = 2.5$; $df = 1981$), for Loblolly/Shortleaf pine, the change is +21.4 m³ ha⁻¹ (+28.8%; $P < 0.05$; 95% CI from +15.9 to +26.9 m³ ha⁻¹; $t = 7.6$; $df = 10{,}170$), and for White/Red/Jack pine, the change is +15.3 m³ ha⁻¹ (+29.4%; $P < 0.05$; 95% CI from +4.5 to +26.1 m³ ha⁻¹; $t = 2.8$; $df = 1280$). The results for natural stands are consistent with the results presented in Table 1 that are derived from all observations aged 1 to 100.

The size of the coefficient on the natural log of lifetime $CO_2$ for natural stands is larger though for the analysis using forests aged 1 to 50 than the analysis for Table 1 using forests aged 1 to 100. For Loblolly/Shortleaf, using only observations aged 1 to 50, the parameter

**Table 1 | Change in predicted wood volume due to elevated $CO_2$ at ages 25/50/75 and due to other episodic phenomena from 1970 to 2015 using observations of naturally regenerated stands aged 1–100**

| Forest Type | | Carbon Fertilization 25 years old Mean | CI | 50 years old Mean | CI | 75 years old Mean | CI | Episodic Phenomena Mean Impact Post-2000 v. Pre-1990 | Number of Observations |
|---|---|---|---|---|---|---|---|---|---|
| White/Red/ Jack Pine | Δ (%) | 23.6 | (9.7, 37.6) | 18.0 | (4.0, 31.9) | 14.4 | (0.4, 28.3) | 21.2 | 2362 |
|  | Δ (m³/ha) | 12.9 | (5.3, 20.6) | 18.3 | (4.1, 32.5) | 26.3 | (0.7, 51.9) | *** | 4980 |
|  | Sig. | | | | | | | | |
| Spruce/ Fir | Δ (%) | 17.4 | (5.0, 29.9) | 13.2 | (0.8, 25.7) | 10.6 | (1.8, 19.4) | 19.2 |  |
|  | Δ (m³/ha) | 5.8 | (1.6, 9.9) | 9.0 | (0.5, 17.5) | 10.0 | (1.7, 18.4) | * | 3582 |
|  | Sig. | | | | | | | | |
| Slash/ Longleaf Pine | Δ (%) | 21.8 | (10.3, 33.3) | 16.6 | (5.0, 28.1) | 13.2 | (1.7, 24.8) | −9.8 |  |
|  | Δ (m³/ha) | 11.2 | (5.3, 17.2) | 17.3 | (5.3, 29.3) | 22.5 | (2.9, 42.1) | ** | 15,188 |
|  | Sig. | | | | | | | | |
| Loblolly/ Shortleaf Pine | Δ (%) | 21.9 | (17.1, 26.8) | 16.7 | (11.8, 21.5) | 13.3 | (8.5, 18.2) | −2.3 |  |
|  | Δ (m³/ha) | 17.5 | (13.6, 21.3) | 31.1 | (22.1, 40.1) | 40.1 | (25.5, 54.6) |  | 9780 |
|  | Sig. | | | | | | | | |
| Oak/ Pine | Δ (%) | 20.7 | (15.2, 26.3) | 15.8 | (10.2, 21.3) | 12.6 | (7.1, 18.1) | −2.5 |  |
|  | Δ (m³/ha) | 12.2 | (9.0, 15.5) | 19.4 | (12.6, 26.2) | 25.1 | (14.1, 36.2) |  | 39,268 |
|  | Sig. | | | | | | | | |
| Oak/ Hickory | Δ (%) | 19.0 | (16.4, 21.5) | 14.4 | (11.9, 16.9) | 11.5 | (9.0, 14.1) | −6.2 |  |
|  | Δ (m³/ha) | 11.2 | (9.7, 12.6) | 16.1 | (13.3, 18.9) | 19.7 | (15.4, 24) | *** | 11,446 |
|  | Sig. | | | | | | | | |
| Oak/Gum/ Cypress | Δ (%) | 22.4 | (16.6, 28.1) | 17.0 | (11.2, 22.8) | 13.6 | (7.8, 19.4) | −2.9 |  |
|  | Δ (m³/ha) | 17.7 | (13.2, 22.3) | 25.3 | (16.7, 33.8) | 33.0 | (19.0, 47.0) |  | 7618 |
|  | Sig. | | | | | | | | |
| Elm/Ash/ Cottonwood | Δ (%) | 16.5 | (10.6, 22.4) | 12.5 | (6.6, 18.4) | 10.0 | (4.1, 15.9) | −11.8 |  |
|  | Δ (m³/ha) | 10.3 | (6.6, 14.0) | 14.4 | (7.6, 21.2) | 16.0 | (6.6, 25.4) | *** | 12,514 |
|  | Sig. | | | | | | | | |
| Maple/Beech/ Birch | Δ (%) | 16.4 | (11.7, 21.0) | 12.4 | (7.8, 17.1) | 9.9 | (5.3, 14.6) | −1.8 |  |
|  | Δ (m³/ha) | 10.2 | (7.3, 13.1) | 13.7 | (8.6, 18.8) | 15.6 | (8.4, 22.9) |  | 16,472 |
|  | Sig. | | | | | | | | |
| Aspen/ Birch | Δ (%) | 19.9 | (15.5, 24.2) | 15.1 | (10.7, 19.4) | 12.1 | (7.7, 16.4) | −18.0 |  |
|  | Δ (m³/ha) | 9.7 | (7.6, 11.8) | 14.6 | (10.4, 18.8) | 17.6 | (11.2, 23.9) | *** | 123,210 |
|  | Sig. | | | | | | | | |
| All Forest Groups | Δ (%) | 20.3 | (15.9, 24.6) | 15.4 | (11.0, 19.8) | 12.3 | (7.9, 16.7) | −8.9 |  |
|  | Δ (m³/ha) | 11.8 | (9.2, 14.4) | 18.4 | (13.2, 23.7) | 23.0 | (14.8, 31.2) | *** |  |
|  | Sig. | | | | | | | | |

Matches were created using observations of naturally regenerated plots with the control being observations from 1968–90 and the treatment being observations from 2000–18. Post-matching, full multivariate-regression analysis was performed and the p value for the $CO_2$ variable was less than 0.01 for each forest group individually and also collectively. Then, the carbon fertilization effect from 1970 to 2015 was analyzed using a one-sided t-test to assess the significance of the change in $CO_2$ exposure from 1970 to 2015 on wood volume at 25/50/75 years of age (i.e., a 25-year-old stand in 1970 received the sum of yearly exposure values from 1946 to 1970 and a 25-year-old stand in 2015 received the sum of yearly exposure values from 1991 to 2015). Climate variables were held at their 1970 levels. The impact of the other episodic phenomena was captured by the dummy variable comparing observations pre-1990 with those post-2000. Using a one-sided t-test to test the hypothesis that volume in 2015 had increased significantly from the volume in 1970, Table 1 displays a 99% confidence interval for all but Spruce/Fir at 75 years of age, where a 95% CI is shown. Model (t) results (Supplementary Data 12–22) were used as inputs in these calculations. ***P < 0.01, **P < 0.05, and *P < 0.10.

**Table 2 | Change in predicted volume from 1970 to 2015 due to carbon fertilization on naturally regenerated and planted pine stands at ages 10 and 25 using observations aged 1–50**

| Age (years) | | Slash/Longleaf | | Loblolly/Shortleaf | | White/Red/Jack | |
|---|---|---|---|---|---|---|---|
| | | Natural | Planted | Natural | Planted | Natural | Planted |
| | Obs. | 2036 | 914 | 10,226 | 2814 | 1334 | 726 |
| 10 | Δ (%) | 24.5 | 24.6 | 32.5 | 30.9 | 33.2 | 48.3 |
| | 95% CI | (10.2, 38.7) | (5.3, 43.9) | (26.3, 38.8) | (19.5, 42.4) | (15.8, 50.6) | (17.1, 79.6) |
| | Δ (m³/ha) | 5.5 | 7.3 | 9.3 | 10.1 | 9.4 | 4.2 |
| | 95% CI | (2.3, 8.6) | (1.6, 13.0) | (7.5, 11.1) | (6.3, 13.8) | (4.5, 14.3) | (1.5, 6.8) |
| 25 | Δ (%) | 21.7 | 21.7 | 28.8 | 27.4 | 29.4 | 42.8 |
| | 95% CI | (7.4, 35.9) | (2.4, 41.0) | (22.5, 35.0) | (15.9, 38.8) | (12.0, 46.8) | (11.5, 74.0) |
| | Δ (m³/ha) | 12.5 | 15.5 | 21.4 | 34.6 | 15.3 | 16.0 |
| | 95% CI | (4.3, 20.7) | (1.7, 29.3) | (16.7, 26.0) | (20.1, 49.1) | (6.3, 24.4) | (4.3, 27.7) |

The data were truncated to observations aged 1 to 50 years. Then matching occurred with control observations spanning 1968–90 and treatment observations spanning from 2000–18. Post-matching, full multivariate-regression analysis was performed and the effect of C fertilization was estimated by comparing the average volume given the age-specific, $CO_2$ exposure for 1970 and 2015 (i.e., a 25-year-old stand in 1970 would have received the sum of yearly exposure values from 1946 to 1970 and a 25- year-old stand in 2015 would have received the sum of yearly exposure values from 1991 to 2015). The p value for the $CO_2$ variable was less than 0.01 for both the natural and planted runs for each forest group. Climate variables were held at their age-specific, 1970 levels. Using a one-sided t-test to test the hypothesis that volume in 2015 was significantly different than volume in 1970, Table 2 displays the 95% confidence interval.

on the carbon-fertilization variable is 1.66 ($P < 0.01$; 99% CI from 1.59 to 1.73; $t = 43.8$), whereas, when using observations aged 1 to 100, the parameter is smaller at 1.29 ($P < 0.01$; 99% CI from 1.25 to 1.33; $t = 59.9$) (Supplementary Table 10). Likewise, for White/Red/Jack pine, the values are 1.70 ($P < 0.01$; 99% CI from 1.49 to 1.90; $t = 16.1$) and 1.38 ($P < 0.01$; 99% CI from 1.26 to 1.50; $t = 21.9$) (Supplementary Table 10), respectively. These larger parameters, when using just the observations of younger stands, suggest that the effect of carbon fertilization could attenuate over time for some forest groups. For Slash/Longleaf though, the parameter values are relatively consistent at 1.25 ($P < 0.01$; 99% CI from 1.08 to 1.42; $t = 14.5$) and 1.27 ($P < 0.01$; 99% CI from 1.16 to 1.37; $t = 23.9$) (Supplementary Table 10), respectively.

## Discussion

Our estimated response of wood volume in ecosystems to elevated $CO_2$ is stronger than that predicted by experimental results. Over the period 1970 to 2015, $CO_2$ concentrations increased by 75 ppm, and we find that this increase in $CO_2$ stimulated an increase in wood volume in naturally regenerated 75-year-old forests in the United States by 12.3% (7.5 and 17.1%: 99% CI). This is larger than the 8.2% effect found in experimental studies[20], although those estimates included non-forested ecosystems, which do not appear to respond as strongly as wood volume to elevated $CO_2$. Also, our results are for temperate forest groups that exist largely in parts of the Eastern US where the climate is mostly warm and wet. This is important given the recent finding that, in settings that are colder and have more water stress, $CO_2$ exposure generates smaller effects on tree-volume[19].

For the southern pine forest groups, our estimates indicate that the proportional response of wood volume in planted stands is roughly equivalent to the response in natural stands, but the results in cubic meters per hectare are larger for planted stands because these planted stands have more volume on average than naturally regenerated stands. For White/Red/Jack pine, our estimates indicate a different pattern. The proportional response of wood volume in planted stands is far larger than the response in natural stands, but the results in cubic meters per hectare are smaller for planted stands early in life. Potentially, there are some management activities associated with the replanting of White/Red/Jack pine, such as competition management, that initially reduce biomass. Using the results for 25-year-old stands (Table 2), the 75-ppm increase in $CO_2$ between 1970 and 2015 generated a 28.8% (21.4 m³ ha⁻¹) increase in wood volume in 25-year-old, naturally regenerated Loblolly/Shortleaf stands and a 27.4%

(34.6 m³ ha⁻¹) increase in planted stands of the same age. This suggests that from 1970 to 2015, wood volume on natural Loblolly/Shortleaf stands increased by about 0.5 m³ ha⁻¹ yr⁻¹ due to elevated $CO_2$, while volume increased by 0.8 m³ ha⁻¹ yr⁻¹ on planted stands.

The parameter on logged $CO_2$ for Loblolly/Shortleaf and White/Red/Jack pine is smaller for the regressions using forests aged 1 to 100 than those using only forests aged 1 to 50 (Supplementary Table 10). This suggests that for these two forest groups, carbon fertilization may attenuate over time—that is, the proportional response in wood volume to elevated $CO_2$, when measured with data including older stands, is smaller than when measured with data including only younger stands. Although the proportional effect of elevated $CO_2$ appears to decline for older stands, the additional accumulation of volume in older stands due to elevated $CO_2$ remains substantial nonetheless because older stands have more wood volume.

The methods in this study explicitly control for other time-linked phenomena. Climate change is controlled directly through our experimental design by matching observations before 1990 with those after 2000 and then including polynomial functions of seasonal climate variables in our post-matching estimations. Other episodic factors, such as nitrogen deposition and invasive species, are controlled with dummy variables that account for the year the plot was observed. While we cannot observe these specific factors, the results indicate that, on average, these episodic factors have caused wood volume in naturally regenerated stands to decline. In fact, across all forest groups, these time-related factors reduced volume by 8.9% ($P < 0.01$) between 1970 and 2015 (Table 1). Only two of the forest groups showed an increasing trend in wood volume that is not attributable to climate change or elevated $CO_2$: White/Red/Jack pine ($P < 0.01$) and Spruce/Fir ($P < 0.10$). We hypothesize that these results are both tied to recovery from past natural disturbances.

For planted stands aged 1 to 50, we find a significant negative trend in wood volume for Loblolly/Shortleaf and White/Red/Jack (but not for Longleaf/Slash) pine that is attributed to episodic temporal factors unrelated to elevated $CO_2$ and climate change, but for similarly aged naturally regenerated stands, the effect is smaller or not significant (Supplementary Data 31–33). This outcome suggests that on planted stands, temporal factors unrelated to elevated $CO_2$ and climate change have had a negative influence on wood volume in the last 30 to 40 years. Although we control for stocking conditions in the matching process, increased thinning activity or lower investments in fertilizing after 2006, as timber prices declined and fertilizer prices

increased during the Great Recession, could explain this result. Forest managers, as well, may have shifted management objectives from being largely focused on increasing volume to increasing value.

Overall, our results present evidence that there has been a positive response in wood volume to elevated $CO_2$ in ten temperate forest groups in the United States. Our matching approach, which allows us to identify the effect of elevated $CO_2$, does not enable us to assess how elevated $CO_2$ interacts with its resulting climate impacts, such as changes in temperature, precipitation, forest composition, and range. The pooled modeling approach we include in the robustness analysis (Supplementary Data 3, 4) may provide an opportunity to test these interactions. Our results hint that the effects of elevated $CO_2$ could attenuate in older forests, but we cannot directly test for attenuation with our approach. Future research should address this question.

## Methods

### Materials
Information on wood volume and the physical environment of the plots were obtained from the US Forest Service Forest Inventory and Analysis (USFS-FIA)[22]. The FIA database categorizes each plot into one of 33 forest groups, but 23 groups do not have sufficient data in the control period (before 1990) to enable robust matching and so were dropped from this study. As a result, several western forest groups (e.g., Douglas-fir) were not included in our study. The following ten forest groups [(1) Loblolly/Shortleaf Pine, (2) Slash/Shortleaf Pine, (3) White/Red/Jack Pine, (4) Spruce/Fir, (5) Elm/Ash/Cottonwood, (6) Maple/Beech/Birch, (7) Oak/Hickory, (8) Oak/Gum/Cypress, (9) Aspen/Birch, and (10) Oak/Pine] all had more than 5000 observations and large numbers of observations both from before 1990 and from 2000 on. Data for the 48 conterminous states from evaluation years between 1968 and 2018 were included in the study. We limited our analysis to plots with trees from 1 to 100 years of age, resulting in trees that had been planted somewhere between 1869 and 2018—a period during which atmospheric $CO_2$ increased from roughly 287 to more than 406 ppm[32–34]. The geographic distribution of the ten forest groups presented in Fig. 2 shows in orange all counties in which the USFS recorded in at least one year between 1968 and 2018 the presence of a plot of the respective forest group that met the age requirements for inclusion in this study. Precipitation and temperature data were obtained from the PRISM Climate Group[41].

### Methods
Results in Tables 1 and 2 are based on estimated exponential tree-volume functions of the generalized form shown in Eq. 1. The left-hand side is the natural log of the volume per hectare in the central stem of trees on each plot in cubic meters. Volume is assumed to be a function of age, the logged cumulative lifetime concentration of $CO_2$, and other variables, including plot-specific variables that vary across plots but not time ($X_i$), weather variables that vary across plots and time ($W_{it}$), and time-specific fixed effects that vary across time but not plots ($E_t$).

$$\text{Ln}\left(\frac{\text{Volume}}{\text{Hectare}}\right)_{it} = \alpha + \beta_0 \frac{1}{\text{Age}_{it}} + \beta_1 \text{Ln}(\text{CumCO2Life}_t) + \beta_2 X_i + \beta_3 W_{it} + \beta_4 E_t + \varepsilon_{it} \quad (1)$$

The nonparametric smearing estimate method was used to transform logged-volume results into a volume in cubic meters per hectare[42]. The climate variables, obtained from the PRISM Climate Group[41] and described in Supplementary Table 1, enter as cubic polynomials of the lifetime seasonal temperature and precipitation averages that a plot of a given age at a given time experienced.

The variable for atmospheric carbon was constructed as the logarithmic transformation of the sum of yearly atmospheric $CO_2$ exposures over the lifetime of the stand. Other site-specific covariates were obtained from the FIA data (Supplementary Table 2), such as the

availability of water, the quality of the soil, the photoperiod of the plot, whether disturbances had impacted the land, and whether the land was publicly or privately owned[43,44].

The time-specific fixed effects ($E_t$) in the model control for episodic factors like nitrogen deposition and invasive species, which are correlated with time but cannot be observed over space for the whole time period. These time-dummy variables account for underlying, unobservable systematic differences between the 21st-century period when atmospheric $CO_2$ was higher and the pre-period when levels were much lower. Controlling for these factors aids the identification of the impact of elevated $CO_2$, which varies annually.

A potential concern is that wood volume changes over time could be related to an increased number of trees per hectare rather than increased wood volume of the trees. To assess whether controls for the stocking condition were needed, we examined data on the number of trees per acre of each forest type. First, we looked at a group of southern states (Supplementary Table 3) and found double-digit percentage changes in tree stocking between 1974 and 2017 for seven of the nine forest groups. However, the changes were mixed, with four having increased tree density and five decreasing tree density. The FIA data do not record the Aspen/Birch forest group as present in these southern states in these evaluations.

Examination of a group of northern states involved a comparison of the average stocking conditions around 1985 with those in 2017. The changes in tree density for these forest types (Supplementary Table 4) were also split with four showing increased stocking and five having less dense stocking. The change for Loblolly/Shortleaf pine was relatively large, with stocking density increasing by 27.2%. Slash/Longleaf was not recorded as present in these states in these evaluations.

Next, we analyzed changes, over the period from around 1985 to 2017, in all states east of the 100th meridian, as those states comprised the bulk of the data in our study (Supplementary Table 5). Results for seven of the ten forest groups showed a less dense composition. Loblolly/Shortleaf pine again was shown to have become more densely stocked, with an increase of 13.2%.

The last check included all of the 48 conterminous states and compared changes in stocking conditions from years around 1985 to 2017 (Supplementary Table 6). Seven of the ten forest groups showed decreased stocking density over time. Not surprisingly (because most Loblolly/Shortleaf is located in the Eastern US), the change in Loblolly/Shortleaf pine density is the same for this check as was shown in the results in Supplementary Table 5. Based on the results from all these comparisons and given that stocking density has changed over time, we controlled for it both in the matching and in the multivariate-regression analysis.

Genetic matching (GM), the primary approach used for this analysis, combines propensity score matching and Mahalanobis matching techniques[45]. The choice of GM was made after initially considering other approaches, such as nearest-neighbor propensity score matching with replacement and a non-matching, pooled regression approach. These three options were tested on the samples for Loblolly/Shortleaf pine and Oak/Hickory, and the regression results are presented in Supplementary Data 3-4.

The results across these different approaches were quite similar, suggesting that the results are not strongly driven by methodological choice. We focused on matching rather than a pooled regression approach to help reduce bias and provide estimates closer to those that would be obtained in a randomized controlled trial. When choosing the specific matching approach, we considered that standard matching methods are equal percent bias reducing (EPBR) only in the unlikely case that the covariate distributions are all roughly normal[46] and that EPBR may not be desirable, as in the case where one of two covariates has a nonlinear relationship with the dependent variable[16]. We also noted that GM is a matching algorithm that at each step

minimizes the largest bias distance of the covariates[24] and that GM has been shown to be a more efficient estimator than other methods like the inverse probability of treatment weighting and one-to-one greedy nearest-neighbor matching[24,47–49]. Additionally, when the distributions of covariates are non-ellipsoidal, this nonparametric method has been shown to minimize bias that may not be captured by simple minimization of mean differences[50]. Lastly, as sample size increases, this approach will converge to a solution that reduces imbalance more than techniques like full or greedy matching[48,51,52]. Given the support that this choice has in the literature, we decided to employ GM to create all the matched data used in this study using R software[53].

Artificial regeneration of forest stands, noted as planting throughout the text is used as the main proxy for the impact of forest management. The other indicator of management activity is what can be described as interventions, which are a range of human on-site activities that the USFS details[22]. We define unmanaged land as stands with natural regeneration and where no interventions occurred on the plot.

To create Table 1, we first excluded all plots on which there had been either planting activities or some type of human intervention. Then, we created treatment and control groups by forming two time periods separated by an intervening period of ten years to ensure a more than a marginal difference between the groups in terms of lifetime exposure to atmospheric $CO_2$. The control period used forest plot data sampled between 1968 and 1990, and the treatment period used forest plots sampled between 2000 and 2018. Note that even though the earlier period contains more years, there are fewer overall observations.

Matches were then made to balance the treatment and control groups based on the following observable covariates: (1) Seasonal Temperature, (2) Seasonal Precipitation, (3) Stocking Condition, (4) Aspect, (5) Age, (6) Physiographic Class, and (7) Site Class. The propensity score was defined as a logit function of the above covariates to generate estimates of the probability of treatment. Calipers with widths less than or equal to 0.2 standard deviations of the propensity score were also employed to remove at least 98% of bias[49].

Balance statistics for the primary covariates are presented in Supplementary Data 1–2 and show a strong balance for all covariates across all forest groups. Thus for each forest group, our sample of plots includes control plots (pre-1990) and treatment plots (post-2000) that are comparable (balanced) in climate and other biophysical attributes.

After trimming our sample using this matching process and obtaining strongly balanced matches, we turned to regression analysis, where we employed Stata software[54]. To confirm that we had the most appropriate model structure, tests of the climate and atmospheric carbon variables were undertaken using various polynomial forms, and the main variable of interest, atmospheric carbon, was tested both using a linear lifetime cumulative $CO_2$ variable and a logarithmic transformation of that variable. Results (Supplementary Data 5–10) show that the climate variables were not improved with complexity beyond cubic form. Moreover, selection tools, like the Akaike and Bayesian information criterion, favored the cubic choice, and so we utilized the cubic formulation throughout this study. Results for the $CO_2$ variable were similar in both sign and significance for the linear and logged form. We use the logged form as it allows easier interpretation of the effect, suppresses heteroscedasticity, and removes the assumption that each unit increase in $CO_2$ exposure will have a linear (constant) effect on volume.

The estimated effect of $CO_2$ exposure for each forest group (Supplementary Data 12–21) was estimated using alternate specifications of the independent variables included in Eq. 1. For each forest type, the Model (1) specification (Eq. 2) is the basis for the results presented in Table 1. The $\beta_0$ coefficient details the impact on the

volume of the main variable of interest, atmospheric carbon.

$$\begin{aligned}\ln\left(\frac{volume}{hectare}\right) = {}& \alpha + \beta_0 \ln(\text{LifetimeCO}_2) + \beta_1 \frac{1}{\text{Age}} + \beta_2 \text{Site Class} \\ & + \beta_3 \text{Seasonal Temperature} + \beta_4 \text{Seasonal Temp}^2 + \beta_5 \text{Seasonal Temp}^3 \\ & + \beta_6 \text{Seasonal Precipitation} + \beta_7 \text{Seasonal Precip}^2 + \beta_8 \text{Seasonal Precip}^3 \\ & + \beta_9 \text{Stocking} + \beta_{10} \text{Disturbances} + \beta_{11} \text{Physiographic Class} + \beta_{12} \text{Aspect} \\ & + \beta_{13} \text{Slope} + \beta_{14} \text{Elevation} + \beta_{15} \text{Latitude} + \beta_{16} \text{Longitude} + \beta_{17} \text{Ownership} \\ & + \beta_{18} \text{Time Dummies} + \beta_{19} \text{Seasonal Vapor Pressure Deficit} \\ & + \beta_{20} \text{Length of Growing Season} + \varepsilon \end{aligned}$$

(2)

After estimating Eq. 2 for each forest type individually (Supplementary Data 12–21), all plots were pooled across forest groups, with additional forest-group dummy variables, to estimate a general tree-volume function (Supplementary Data 22).

Our main Model (1) results are provided in Supplementary Data 12–22, along with three additional models that assess the robustness of the elevated $CO_2$ effect to different specifications. The simplest specification, Model (4), included only stand age, $CO_2$ exposure, and a time-dummy variable. Model (3) took the Model (4) base and added in an array of site-specific variables, including those for the climate. Model (2) was similar to Model (1) in that it included the impact of vapor pressure deficit and the length of the growing season on the variables included in Model (3), but it differed from Model (1) in that it tested an alternate approach to capturing the impact of underlying, unobservable systematic differences like nitrogen deposition.

Using the estimated coefficients from the preferred Model (column 1) specification (Eq. 2), the estimated change in growing-stock volume between two $CO_2$ exposure scenarios was calculated at ages 25, 50, and 75. The first scenario examined $CO_2$ exposure up to 1970 (that is, when calculating growing-stock volume for a 25-year-old stand, the $CO_2$ exposure would have the summation of the yearly values for the years from 1946 to 1970 [310 to 326 ppm $CO_2$]). The second scenario examined $CO_2$ exposure up to 2015 (that is, when calculating growing-stock volume for a 25-year-old stand, the $CO_2$ exposure was the summation of the yearly values for the years from 1991 to 2015 [347 to 401 ppm $CO_2$])[32–34]. In both scenarios, climate variables were maintained at their 1970 exposure levels, covering the same historical years (e.g., for a 25-year-old stand, 1946 to 1970 were the years of interest), while using seasonal, not annual values and calculating average values, not lifetime summations.

Forest dynamics in the Western US differ from those in the East (e.g., generally drier conditions; greater incidence of large wildfires) and as most of the observations for this study are of forest groups located in the 33 states that the USFS labels as comprising the Eastern US, robustness tests were conducted to assess whether results would differ were only eastern observations utilized. Three forest groups [(1) Loblolly/Shortleaf pine, (2) Oak/Gum/Cypress, and (3) Slash/Longleaf pine] have no observations in the Western US. A fourth, White/Red/Jack Pine, has a slight presence in a few Western states, but no western observations were selected in the original matching process (Supplementary Data 2). For the other six forest groups, all observations from Western US states were dropped. As can be seen from Fig. 2, this had the biggest impact on Aspen/Birch and Elm/Ash/Cottonwood. With this data removed, the GM matching algorithm was again used. Balance statistics are presented in Supplementary Data 23 and again show a strong balance for all covariates across all forest groups. With matches made, the average treatment effect on the treated was estimated using the Model (1) specification used to create Table 1. Regression results are presented in Supplementary Data 24, 25, and a revised version of Table 1 for just the observations from the Eastern US is presented as Supplementary Table 7.

As an additional robustness check on the results in Table 1, we tested an alternative functional form of the volume function. This

alternative volume function is shown in Eq. 3. It has a similar shape as the function used for the main results in the paper, however, this equation cannot be linearized with logs in a similar way. Thus, it was estimated with nonlinear least squares, using the matched samples of naturally regenerated forests for individual forest groups, as well as the aggregated sample.

$$\frac{\text{Volume}}{\text{Hectare}} = a/(b + \exp(-c * \text{Age})) \tag{3}$$

We began by estimating two separate growth functions, one for the pre-1990 (low $CO_2$) period and one for the post-2000 (high $CO_2$) period using Eq. 3. That is, observations from the pre-1990 (low $CO_2$) control period and from the post-2000 (high $CO_2$) treatment period were handled in separate regressions. For this initial analysis with the nonlinear volume function, we did not control for $CO_2$ concentration or other factors that could influence volume across sites (e.g., weather, soils, slope, aspect), and thus, results likely show the cumulative impact of these various factors. Using the regression results (Supplementary Data 26), we calculated the predicted volume for the pre-1990 and post-2000 periods and compared the predicted volumes (Supplementary Table 8).

Next, we tested this yield function on the combined sample (containing both control and treatment observations) and all forest groups. Here the model was expanded to better identify the impact of elevated $CO_2$ by including all covariates. Instead of using a dummy variable for each forest group, though, a single dummy variable was used to differentiate hardwoods from softwoods. Once again, the equation was logarithmically transformed for ease of comparison with the results presented in Table 1. All covariates were originally input, but those which were not significant were removed. That process yielded the functional form shown in Eq. 4. Results for the regression are presented in Supplementary Data 27. The predicted change in volume due to $CO_2$ fertilization from 1970 to 2015 is shown in Supplementary Table 9.

$$
\begin{aligned}
\frac{\text{Volume}}{\text{Hectare}} = & (a0 + a1 * \text{Time Dummy} + a2 * \text{Ln(LifetimeCO2)} + a3 * \text{Ln(Seasonal Temperature)} \\
& + a4 * \text{Ln(Seasonal Precipitation)} + a5 * \text{Site Class} \\
& + a6 * \text{Physiographic Dummy} + a7 * \text{Aspect Dummy} + a8 * \text{Stocking Code} \\
& + a9 * \text{Disturbances} + a10 * \text{Hardwood/Softwood Dummy})/(b0 + b1 * \text{Time Dummy} \\
& + b2 * \text{Ln(Lifetime } CO_2) + b3 * \text{Ln(Seasonal Temperature)} \\
& + b4 * \text{Ln(Seasonal Precipitation)} + b5 * \text{Site Class} \\
& + b6 * \text{Physiographic Dummy} + b7 * \text{Aspect Dummy} + b8 * \text{Stocking Code} \\
& + b9 * \text{Disturbances} + b10 * \text{Hardwood/Softwood Dummy} \\
& + \exp(-(c0 + c1 * \text{Time Dummy} + c2 * \text{Ln(Lifetime } CO_2 \\
& + c3 * \text{Ln(Seasonal Temperature)} + c4 * \text{Ln(Seasonal Precipitation)} + c5 * \text{Site Class} \\
& + c6 * \text{Physiographic Dummy} + c7 * \text{Aspect Dummy} + c8 * \text{Stocking Code} \\
& + c9 * \text{Disturbances} + c10 * \text{Hardwood/Softwood Dummy}) * \text{Age}))
\end{aligned}
\tag{4}
$$

As the results using the nonlinear volume functions were similar in sign and magnitude to the multivariate-regression results and as the practice of matching and then running a multivariate-regression represents a doubly robust econometric approach that has been shown to yield results that are robust to misspecification in either the matching or the regression model[47,55–57], the main text results are based on estimations utilizing multivariate-regression analysis post-matching.

To develop Table 2, which compares naturally regenerated stands with planted stands, we used the same general approach as was used to create Table 1. The analysis and comparison of planted and naturally regenerated stands was conducted only for stands with enough observations of both to make a comparison: White/Red/Jack, Slash/Longleaf, and Loblolly/Shortleaf pine. We followed the same matching and regression procedures as above, but conducted the matching separately for naturally regenerated and planted stands. We also

limited the data to stands less than or equal to 50 years of age, as there are few planted stands of older ages due to the economics of rotational forestry[35–40]. Balance statistics for the matched samples are presented in Supplementary Data 28–30. Again, the matching process resulted in a good balance in observable plot characteristics, which implies that we achieved comparable treatment and control plots.

Using the matched data, we estimated the same regression as in Eq. 2. Estimation results, which use the Model (2) specification from Supplementary Data 19–21 that was used with the data for these three forest groups from ages 1–100, are presented in Supplementary Data 30–32. A comparison of the parameter estimates on the natural log of lifetime $CO_2$ exposure between the results for ages 1–50 (from Supplementary Tables 31–33) and those for ages 1–100 (from Supplementary Data 19–21) is presented in Supplementary Table 10.

### Reporting summary
Further information on research design is available in the Nature Research Reporting Summary linked to this article.

## Data availability
The compiled data generated from the raw data in the USFS-FIA and PRISM group databases have been deposited in the Ohio State University repository, which is a public database that does not require special permission to access. The data can be accessed using: Click Link [https://u.osu.edu/forest/co2-fertilization/]. Source data are provided with this paper.

## Code availability
The code files have also been deposited in the Ohio State University repository, which is a public database that does not require special permission to access. The code can be accessed using: Click Link [https://www.dropbox.com/sh/c08s1spc4xutw6i/AABF_l-YVfy1rCk12-vqeM3za/FIA_CO2FertilizationData?dl=0&subfolder_nav_tracking=1].

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

## Acknowledgements

Funding for this project was obtained from the USDA National Institute for Food and Agriculture Grant 20176702326275. The findings and conclusions in this publication are those of the author(s) and should not be construed to represent any official USDA or US Government determination or policy. This research was supported [in part] by the US Department of Agriculture, Economic Research Service.

## Author contributions

Conceptualization, Methodology, and Writing—review and editing: B.S., E.C.D., and D.J.L. Data curation: E.C.D. Investigation: E.C.D. Visualization: E.C.D. and B.S. Funding acquisition: B.S. and D.J.L. Project administration: B.S. Software: E.C.D. Supervision: B.S. Writing—original draft: E.C.D. and B.S.

## Competing interests

The authors declare no competing interests.
