## [Peer Review File · Nature Communications]

Reviewers' Comments:

Reviewer #1:

Remarks to the Author:

This paper deploys a clever statistical matching technique to specifically examine the impacts of CO₂ fertilization on forest growth in the eastern US. They match stands of similar ages, site classes, temperature, precipitation histories etc, but different cumulative CO₂ exposure because data derive from 1968 to 1990 (control) versus 2000 to 2018 (treatment). Although dynamic global vegetation models similarly document increased growth with CO₂ fertilization, this demonstration with empirical data is novel, creative and informative.

I reviewed an earlier version of this work for a different journal and find it much improved. Specifically, the inclusion of cumulative CO₂ exposure as a predictor in the model alleviates my prior concerns.

This matching technique is really innovative and its clear that they thought very carefully about how to match the control and treatment plots (Table S6 through S8). I don't have the expertise to evaluate the appropriateness of using time dummies to capture time-varying factors like nitrogen deposition and invasive species. This is an important methodological step, so I recommend another reviewer specifically checks this approach.

My only remaining question is about holding climate variables at their 1970 levels (line 143). Given that both temperature and precipitation are changing with rising CO₂ levels, this means that matched sites might differ in cumulative CO₂ exposure AND temperature/precipitation regimes if seasonal climate variables are held at 1970s level. But from the supplementary material (Table S5) it looks like the authors "calculated the lifetime average seasonal temperature and precipitation for each stand age." In the supplement (lines 649-654), the authors again describe holding seasonal climate averages at ~1970 levels which makes sense for drawing the yield curve for the low-CO₂ period, but what was done for the high-CO₂ model? Please clarify the methods throughout about how seasonal climate was incorporated in the models.

MINOR SUGGESTIONS/QUESTIONS

The authors describe this as an analysis of U.S. forests, but state that the plots are "predominantly in the Eastern US". I did not see a specific breakdown of data coverage in the East versus the West - what does "predominantly" even mean? However, given very different forest dynamics in the West under climate change (e.g., more frequent and severe wildfires that will undermine a CO₂ fertilization benefit) it would be advisable to clarify that this is really an analysis of eastern US forests.

Line 31 (typo): per-hectare volume of trees forests

Line 123: Why are 75-year old stands highlighted here? As the authors point out a key question is whether CO₂ fertilization attenuates in older stands (line 78), so the fact that the delta from CO₂ fertilization declines from 25 to 50 to 75 (Table 1) is interesting and worth making more obvious in the text.

Figure 1/Table 2: Is it possible to add error to these curves/estimates? Table 1 includes 99% CI, which are very helpful and informative. Also, Figure S2?

Tables S9-S18: It would be helpful to clarify in the legend the different models. Also, it wasn't clear what models 2/3 test beyond the "robustness to different time dummy variable specifications". Please clarify methods here. Also, why are there separate rows for mean spring temperature etc? there are little "2" and "3" footnote indicators, but no footnotes. Please clarify methods here.

Tables S19 and S20: Given that the comparison is between naturally regenerated versus planted stands, I think it would be more helpful to have put the naturally regenerated and planted balance statistic summaries for each forest type together (e.g., just white/red/jack pine). That makes it

easier to see that – for example – the natural regeneration plots have slightly higher spring precipitation and lower summer precipitation in the white/red/jack pine natural regeneration compared to planted stands.

Reviewer #2:

Remarks to the Author:

The article "The Role of Carbon Fertilization on Naturally Regenerated and Planted U.S. Forests" presents a statistical analysis of the effect of CO₂ fertilization on forests in the US, using an extensive forest inventory and climate dataset. The article itself is well written and the topic of substantial interest to the scientific community. I have to be honest that I am having a hard time with reviewing the manuscript though, in particular as there is not even a single plot showing data (rather an almost trivial plot of predicted fertilization, which really doesn't add information). Looking at table S6 and S7 and the description of the analysis methods, I have a couple of questions that I would need to really judge the analysis:

- 1) Why did the authors choose a log(CO₂), instead of a linear CO₂ dependence? If they tried, what was the difference?
- 2) Temperature datasets: Did the authors try different metrics such as growing degree days (often used for crop yields)? The choice made here seems rather limited, as growth often responds to extremes (e.g. nighttime minimum temperatures) rather than the mean state. This analysis could be expanded to be honest. There are plenty of options, i.e. length season of non-freezing nights, 95th percentile temperature summer extremes, etc.
- 3) No measure of humidity such as VPD or relative humidity is included?
- 4) As CO₂ can almost be seen as a near-linear proxy of time, did the authors take into account changes in NO₂ emissions of the investigated state? Especially the eastern US can be highly impacted.
- 5) The R² overall seems pretty poor, indicating that either the model itself is not capturing all underlying drivers of that the data is noisy. Can you judge (and show) which one of these options is the reason for the low R²? If it is not noise, I wouldn't trust the confidence levels.
- 6) How would the R² change if you remove CO₂ from the fit? A plot showing the data (even if aggregated to reduce the noise) would really help getting a feel for data that is being used.

A few minor issues:

Line 17: "This paper uses empirical data" What is empirical data? (I assume you just mean empirical analysis?)

Line 79: "by applying cutting-edge econometric research design" what is so cutting edge about it? Usually, my alarm bells go off if I read these words, so they can do more harm than good, state-of-the-art or modern would be just fine.

As for your matching methods: This must have been a tedious exercise and I applaud the authors for it. I would be curious though how much "worse" the analysis and fits would have been without matching (or slightly different matching metrics)? It would be important to show that the results don't depend on ad-hoc matching choices.

Reviewer #3:

Remarks to the Author:

This paper uses FIA plot data to determine the CO₂ fertilization effect on timber volumes. It uses a unique and robust statistical approach to quantify the % effect of CO₂ on volume and exclude confounding factors.

The supplementary materials are well documented. The authors have done a great job providing a concise summary of a complex topic.

Reviewer #4:

Remarks to the Author:

General comments

Based on great number of plots data from USFS-FAI, the authors created treatment and control groups by forming two time periods and balanced the plots of two groups by GM matching algorithm. Then, they developed multivariate regression models with dummy variables, and analyzed the effects of elevated CO₂ on volume growth and compared the difference of effects between planted and natural stands. Focusing on the modeling and dummy variables used by the authors, it seems to be sound. The data were obtained from the USFS-FAI database, and much enough for modeling. The two groups were classified appropriately, and the best matching algorithm was used for balancing the plots. The results were presented in sufficient detail, and analyzed reasonably. However, the conclusion "the 75-ppm increase in CO₂ between 1970 and 2015 engendered a 29.6% to 35.7% increase in aboveground biomass in naturally regenerated stands and a 33.4% to 36.6% increase in planted stands" really surprised me. On my opinion, the model structure might need to be taken into consideration again.

This paper aims to identify the impact of elevated CO₂ on volume growth, then a base growth function is necessary. On line 389, the authors stated to use a set of logistic growth functions, but the three models (1), (2) and (3) with dummy variables and the model (4) without dummy variables are not really growth functions, they are multivariate regression models. For a growth model, the key variable is age, other variables are less important, which can be used as dummy variable or random variable affecting the parameters of growth function. For a multivariate regression model, all variables are equally important, and affect independently the target variable such as volume. This is the difference between two kinds of models.

In this study, all the variables in equations S1 and S2 would affect the estimate of volume, not volume growth. Age is not the key variable, which affect the volume estimate together with other variables. To analyze the effect of elevated CO₂ on volume growth, we need to develop two average growth functions based on the data of treatment and control groups, and analyze whether the difference is significant or not. Also, we can develop one average growth function with a time-dummy variable.

Taking the logistic function as an example, the average volume growth function is as follows:

$$V = a / (b + \exp(-c * \text{Age}))$$

And the general volume growth function with dummy variable is as follows:

$$V = (a_0 + a_i * X_i) / ((b_0 + b_i * X_i) + \exp(-(c_0 + c_i * X_i) * \text{Age}))$$

X_i can be defined as all variables of interest, such as elevated CO₂ and climate factors. If some of the a_i , b_i , and c_i are not significant, then delete them and refit the model again. In addition, the volume data exhibit heteroscedasticity, that is, the error variance is not constant over all observations. Thus, weighted regression or logarithmic regression should be used for parameter estimation of the growth models.

For the purpose of developing a multivariate regression model, the GM matching algorithm should be efficient. But for developing a growth model, I can't make sure whether the balanced plots of two groups are sufficiently matched, because age is the most important variable.

Finally, it is suggested find more references about growth models.

Special comments

Line 7, 18, 22-23, 134, 173, 190, 239, 250, 534, 709, and 712: "biomass" needs to be changed to "volume", because the target variable in equation S1 or S2 is volume, not biomass.

Line 125: "between 11 and 15% (12 to 41 m³ha⁻¹)" need to be changed to "between 10.7% and 15.2% (12.0 to 40.7 m³ha⁻¹)", keeping them consistent with the values in Table 1.

Line 129: "+13% (+26 m³ha⁻¹)" need to be changed to "12.9% (25.6 m³ha⁻¹)", keeping them consistent with the values in Table 1.

Table 1: All of the values in brackets need to keep 1 decimal digit, for example, (1.4, 29) need to be changed to (1.4, 29.0). That is, the zero ".0" can't be omitted.

Line 186: "24%, 29% and 24%" need to be changed to "24.0%, 29.0% and 24.3%", keeping them consistent with the values in Table 2.

Line 188: "27%, 30% and 33%" need to be changed to "27.1%, 29.8% and 32.7%", keeping them consistent with the values in Table 2.

Line 208: What does DGVM mean? The detailed expression should be given.

Line 224: We can't find 14.4% in all tables, which need to be provided in the result section.

Line 240-242: It should be pointed out it is only for 75 years old forests, and "13% (+/- 2%; p<0.01)" need to be changed to "12.9% (between 11.3% and 14.5% for p<0.01)", keeping them consistent with the values in Table 1.

Line 362, 718-719: We can't find references numbered as 40 and 41. I guess the references 42 and 43 to be 40 and 41, respectively.

Line 454-456: Six decrease and three increase, not five and four.

Line 562-628: In the tables S9-S18, the estimates of parameters should keep the same significant digits, not same decimal digits. For example, the range of significant digits for parameter estimates of model (1) in table S9 is from 1 (0.000004 for mean winter precipitation) to 9 (190.999293 for constant), this is not the scientific expression. The best way is to use scientific notation. For example, the five parameter estimates in the most front are expressed as 1.103, 5.219, 1.836, -3.212×10^{-1} , 1.620×10^{-2} , respectively. In general, 4 or 5 significant digits are enough. Same as in tables S21 and S22.

Line 668: Figure S2 should be Figure S1.

REVIEWER COMMENTS

Reviewer #1 (Remarks to the Author):

This paper deploys a clever statistical matching technique to specifically examine the impacts of CO₂ fertilization on forest growth in the eastern US. They match stands of similar ages, site classes, temperature, precipitation histories etc, but different cumulative CO₂ exposure because data derive from 1968 to 1990 (control) versus 2000 to 2018 (treatment). Although dynamic global vegetation models similarly document increased growth with CO₂ fertilization, this demonstration with empirical data is novel, creative and informative.

I reviewed an earlier version of this work for a different journal and find it much improved. Specifically, the inclusion of cumulative CO₂ exposure as a predictor in the model alleviates my prior concerns.

This matching technique is really innovative and its clear that they thought very carefully about how to match the control and treatment plots (Table S6 through S8). I don't have the expertise to evaluate the appropriateness of using time dummies to capture time-varying factors like nitrogen deposition and invasive species. This is an important methodological step, so I recommend another reviewer specifically checks this approach.

My only remaining question is about holding climate variables at their 1970 levels (line 143). Given that both temperature and precipitation are changing with rising CO₂ levels, this means that matched sites might differ in cumulative CO₂ exposure AND temperature/precipitation regimes if seasonal climate variables are held at 1970s level. But from the supplementary material (Table S5) it looks like the authors "calculated the lifetime average seasonal temperature and precipitation for each stand age." In the supplement (lines 649-654), the authors again describe holding seasonal climate averages at ~1970 levels which makes sense for drawing the yield curve for the low-CO₂ period, but what was done for the high-CO₂ model? Please clarify the methods throughout about how seasonal climate was incorporated in the models.

We agree that temperature and precipitation are changing with CO₂ but have controlled for that by matching between the post-2000 period and the pre-1990 period on both temperature and precipitation, as well as other variables. We use lifetime average seasonal temperature and precipitation, which includes information about age, which we also use in the matching (see Lines 675-689 as reference). So our matched sites should have similar lifetime average temperatures, precipitation levels, and similar ages, but since they came from two different time periods they will have different exposures to CO₂.

When calculating the impact of carbon fertilization, for both the low and high CO₂ periods, we held seasonal climate averages at their 1970 levels in order to isolate the role of CO₂ exposure. Not doing so would have allowed for changes to be also due to temperature and precipitation. By controlling in this way, the only factor driving the changes we showed in the figure and calculations was carbon fertilization, which is the impact that we wanted to highlight. We are, however, removing this figure based on comments from other reviewers.

MINOR SUGGESTIONS/QUESTIONS

The authors describe this as an analysis of U.S. forests, but state that the plots are “predominantly in the Eastern US”. I did not see a specific breakdown of data coverage in the East versus the West – what does “predominantly” even mean? However, given very different forest dynamics in the West under climate change (e.g., more frequent and severe wildfires that will undermine a CO₂ fertilization benefit) it would be advisable to clarify that this is really an analysis of eastern US forests.

We appreciate the reviewer's question and have tried to be more precise on this point. To that end, we have taken several steps. First, we included Figure S1 (Line 609) (that shows the range of each forest group based on the observations taken by the USFS. We have also dropped any reference to predominantly eastern forests (e.g. Lines 11, 121, and 130). Finally, we have conducted a robustness check where all observations that were not in the 33 states that the USFS defines as part of the Eastern U.S. were dropped (Lines 779-813). This resulted in the number of observations in our matched data being reduced by just over one percent (Tables S28-S29). Then, the same regressions were run (Tables S30-S31) and Table 1 recreated as Table S32 in the supplement. The results differ only modestly.

Line 31 (typo): per-hectare volume of trees forests
Typo has been fixed (Line 23).

Line 123: Why are 75-year old stands highlighted here? As the authors point out a key question is whether CO₂ fertilization attenuates in older stands (line 78), so the fact that the delta from CO₂ fertilization declines from 25 to 50 to 75 (Table 1) is interesting and worth making more obvious in the text.

We chose the 75-year old stand because the volume weighted average age of the species we examined is above 75 years (~90). However, when matched, we end up with relatively few plots above 75 years of age (because the pre-1990 forest was younger), so we limit our assessment in Table 1 to 75 years.

The delta from CO₂ effects in this approach that is noted is due to the fact that older stands spend proportionally more years at lower levels of CO₂ (Lines 141-143 and 417-425) than younger stands (e.g., in 2010, a 75 y-o stand spent 35 years before 1970 while a 50-year old stand spent only 10 years, and a 25 y-o stand spend no years before 1970, when CO₂ levels were substantially lower). So for instance between 1970 and 2015, cumulative lifetime CO₂ increased 11.1% for a 75 y-o stand, whereas for a 25 y-o stand, it increased 18.9%. This is not attenuation but instead is the effect of a larger proportional increase in CO₂ on younger stands, and so the constant CO₂ effect we estimate multiplied by a larger proportional change in CO₂ for younger stands leads to a larger proportional change in biomass. Note that the change in biomass itself is larger for older stands, so the increase in biomass is bigger for older stands, even though the proportional change is not.

However, we do have some evidence of attenuation from the later comparison of planted and natural stands, for which we used only stands aged 1 to 50 (because there are so few planted stands that are older than 50 years due to the intensive management practices) (Lines 469-476 and 856-857). The parameter on carbon fertilization is larger for the younger stands, which suggests that the effect of fertilization may attenuate for older stands, with the larger sample including observations from older

stands, which reduces the average effect of fertilization. We have included some discussion of this in the text (Table S39).

Figure 1/Table 2: Is it possible to add error to these curves/estimates? Table 1 includes 99% CI, which are very helpful and informative. Also, Figure S2?

Thanks. Good point. A CI has been added to Table 2 (Line 181).

Tables S9-S18: It would be helpful to clarify in the legend the different models. Also, it wasn't clear what models 2/3 test beyond the "robustness to different time dummy variable specifications". Please clarify methods here.

Thanks for this. We have clarified the table headings. Model 1 includes our preferred specification that contains the most stringent climate controls but only a single time dummy to capture the impact of unobserved correlates that change over time, such as nitrogen deposition, bug infestations, or other widespread phenomena. Model 2 does indeed test the robustness to a different time dummy variable specification. With both of these models, these variables are crucial as they allow us to identify the CO₂ effect. Model 3 looks at the impact of Vapor Pressure Deficit and growing season on the overall model. Model 4 tests a stripped-down model that contains only CO₂ concentration, age, and time dummy variables. These have been clarified in the legend as requested (e.g. Lines 702-705).

Also, why are there separate rows for mean spring temperature etc? there are little "2" and "3" footnote indicators, but no footnotes. Please clarify methods here.

Thanks for pointing this out. The rows are for the climate variables specified as a squared or cubed term in a polynomial. Thus, the numbers represented squared and cubic terms and not footnotes. We have written this out now (e.g. Table S7 on Line 646).

Tables S19 and S20: Given that the comparison is between naturally regenerated versus planted stands, I think it would be more helpful to have put the naturally regenerated and planted balance statistic summaries for each forest type together (e.g., just white/red/jack pine). That makes it easier to see that – for example – the natural regeneration plots have slightly higher spring precipitation and lower summer precipitation in the white/red/jack pine natural regeneration compared to planted stands.

Thanks. We agree. The tables were changed as suggested (Tables S33-S38 on Lines 817-855).

Reviewer #2 (Remarks to the Author):

The article "The Role of Carbon Fertilization on Naturally Regenerated and Planted U.S. Forests" presents a statistical analysis of the effect of CO₂ fertilization on forests in the US, using an extensive forest inventory and climate dataset. The article itself is well written and the topic of substantial interest to the scientific community. I have to be honest that I am having a hard time with reviewing the manuscript though, in particular as there is not even a single plot showing data (rather an almost trivial plot of predicted fertilization, which really doesn't add information).

Thanks for this suggestion. We have added a new figure (Fig. 1 on Lines 29-31) to the main text which shows the data on growing-stock volume for two years (1997 and 2017) and for each forest group that we analyze. This new figure also sets up our key research question of studying why per-acre volumes have increased over time. We have also included a figure (Figure S1 in the supplement on Lines 609-610) that shows the range for each forest group based on the observations that were taken by the USFS that comprise the data used in this study.

Looking at table S6 and S7 and the description of the analysis methods, I have a couple of questions that I would need to really judge the analysis:

1) Why did the authors choose a log(CO₂), instead of a linear CO₂ dependence? If they tried, what was the difference?

We tested using a linear CO₂ variable before ultimately choosing the logged variable approach. However, given this comment, we now present results from a linear model in Tables S14 and S16 of the supplement (Lines 690-692 and 696-698). The parameter on CO₂ is positive and significant, although smaller in absolute value. In the linear form, each 1 unit increase in CO₂ is assumed to have a constant percentage effect on volume. This specification creates the result that for higher levels of CO₂ in the atmosphere, a 1% increase in CO₂ has a larger effect on volume than a 1% increase from lower levels of CO₂ in the atmosphere. The logged model that we use (Tables S13 and S15: Lines 687-689 and 693-695) was preferred as it is more conservative and generates elasticity interpretations where each percentage change in CO₂ has a constant percentage change in volume. Plus, a comparison of the R² metrics between the linear and logged models shows that the log model fits the data better.

2) Temperature datasets: Did the authors try different metrics such as growing degree days (often used for crop yields)? The choice made here seems rather limited, as growth often responds to extremes (e.g. nighttime minimum temperatures) rather than the mean state. This analysis could be expanded to be honest. There are plenty of options, i.e. length season of non-freezing nights, 95%ile temperature summer extremes, etc.

Thank you for your comments on modeling temperature. We agree that there are quite a number of different variables that could be used. One of our main constraints was trying to find a form that was flexible enough to fit all the forest groups that are covered in this study, as they don't respond uniformly to climate or climate change. Our choice of a polynomial with squared and cubed terms for temperature and precipitation, we believe, allows for differential effects across the observed ranges of those variables, while maintaining a uniform approach for all forest groups.

In response, however, we have tested additional variables, such as the number of growing months per year that a stand experienced across its lifetime, and we have also used variables like the cumulative number of months where the mean monthly temperature was above a specified threshold (30C or 35C).

These results are presented in the supplementary information packet in Tables S13 through S16: Lines 687-698, however, they do not appreciably change our estimate of primary interest, which is the parameter on logged cumulative lifetime CO₂.

3) No measure of humidity such as VPD or relative humidity is included?

This is a great point. We have downloaded the data from the PRISM database and tested it using both annual and seasonal averages. As the results showed this to be a highly salient variable, we have included it in our main regression runs. That said, the inclusion of vapor pressure deficit had a very small impact on the carbon fertilization parameter, changing it from 1.134 to 1.158 for the combined forest group run (Table S27: Lines 772-777).

4) As CO₂ can almost be seen as a near-linear proxy of time, did the authors take into account changes in NO₂ emissions of the investigated state? Especially the eastern US can be highly impacted.

We agree that N deposition is important. Unfortunately, there is not a database that we know of that tracks spatially across the same region as our data for the full period of observation. To overcome this limitation, we include time-dummy variables that are created in different ways (Table S2: Lines 623-624). One formulation defines the time dummy as equal to one for USFS evaluations conducted in the 21st century (when N deposition was generally lower) and equal to zero for the base period of such evaluations between 1968 and 1990. We also try to further aid the identification by separately defining the time dummies for each decade for which we have observations, e.g. a 1980s dummy variable equals 1 if the observation was in the 1980s and equals 0 otherwise (e.g. Table S17: Lines 700-705). This captures trends in the outcome (ln(volume) over time, but does not allow us to separately identify N deposition from other trends, such as new invasives or other episodic events. However, note that any episodic events that are directly controlled by climate will be controlled by our approach, which accounts for temperature and precipitation changes.

5) The R² overall seems pretty poor, indicating that either the model itself is not capturing all underlying drivers of that the data is noisy. Can you judge (and show) which one of these options is the reason for the low R²? If it is not noise, I wouldn't trust the confidence levels.

Thank you for this comment. We are taking the quasi-experimental econometric perspective with this research and are attempting to achieve a consistent estimate of CO₂ on growing stock volume, which is different than using regression to predict volume. Therefore, our research strategy is designed to guard against omitted variable bias in terms of our CO₂ estimate through matching parcels across time based on observed covariates, and through post-matching regression that includes variables that might be correlated with CO₂ (e.g. climate, time dummies). Our biggest omitted variable bias concerns revolve around climate variables and episodic shocks. Here is a summary of how we think about the research design generating consistent estimates of the CO₂ impacts:

- Climate variables impact growing stock volume and are likely correlated with our CO₂ exposure variable since climate has changed over the time span of our data. Thus, we match pre-1990 plots with post-2000 plots based on many covariates, including climate. The result of the matching process is a balanced sample (Tables S9-S12: Lines 660-677), whereby the distribution of climate is similar across the pre-1990 and post-2000 plots. To ensure there is no remaining bias, we also include the climate variables in the set of independent variables for the post-matching growing-stock volume regressions.

- We are concerned that there are large-scale episodic shocks (like pests) that impact growing-stock volume and have changed over time such that they are correlated with CO₂ exposure. Therefore, we include time dummy variables (equal to 1 for plots observed in particular periods like the 1970s, and equal to 0 otherwise) to capture such episodic shocks. Since plots of different ages that are observed in the same time period have different lifetime CO₂ exposures, we are able to include both CO₂ exposure and time dummies in the same post-matching regression, which then guards against omitted variable bias from episodic shocks.

6) How would the R² change if you remove CO₂ from the fit? A plot showing the data (even if aggregated to reduce the noise) would really help getting a feel for data that is being used.

As requested, we have included tables that show how the addition/removal of CO₂ impacts the value of R² in Tables S40 and S41: Lines 881-891. As expected, R² is lowest when only age is included. Including CO₂ then helps explain volume, which makes sense because the data are plots of a bunch of ages matched to plots of the same ages in an earlier time period. When other variables are included to explain biomass, but CO₂ is not included, R² is higher than when only CO₂ fertilization is used. Finally, when all variables are included, R² is greatest, as expected. When comparing the last two models (with and without CO₂), the parameters that are fixed over time but vary across space remain fairly consistent, while the parameters on variables that shift over time and space vary a lot between the two models. They vary because they are correlated with the CO₂ variable, meaning that the model without CO₂ produces biased estimates on the climate variables as well as other variables that vary across time because of the exclusion of CO₂.

A few minor issues:

Line 17: "This paper uses empirical data" What is empirical data? (I assume you just mean empirical analysis?)

Thank you for catching that. Changed as suggested (Line 10).

Line 79: "by applying cutting-edge econometric research design" what is so cutting edge about it?

Usually, my alarm bells go off if I read these words, so they can do more harm than good, state-of-the-art or modern would be just fine.

We appreciate your words of caution. Text has been revised using your suggestion (Line 77).

As for your matching methods: This must have been a tedious exercise and I applaud the authors for it. I would be curious though how much "worse" the analysis and fits would have been without matching (or slightly different matching metrics)? It would be important to show that the results don't depend on ad-hoc matching choices.

As requested, we include tables (Tables S7-S8: Lines 645-656) showing the results for two of the ten forest groups using alternative methodologies. We compare two forms of matching with a regression analysis that does not involve matching. That is, the multivariate regression is run on the full sample of unmatched data. The results illustrate the importance of using matching. For the softwood type, the unmatched sample underestimates the effect of carbon fertilization, while for hardwoods there is not much of an impact associated with using the matched versus unmatched data. In spite of these similar results, we chose a matching approach, as this is known to create data that more closely mimics that of a randomized controlled trial and thus more clearly identifies the impact of the key variable of interest, which here is carbon fertilization, by reducing sources of confounding bias.

Reviewer #3 (Remarks to the Author):

This paper uses FIA plot data to determine the CO₂ fertilization effect on timber volumes. It uses a unique and robust statistical approach to quantify the % effect of CO₂ on volume and exclude confounding factors.

The supplementary materials are well documented. The authors have done a great job providing a concise summary of a complex topic.

We thank the reviewer for the kind words about our manuscript.

Reviewer #4 (Remarks to the Author):

General comments

Based on great number of plots data from USFS-FAI, the authors created treatment and control groups by forming two time periods and balanced the plots of two groups by GM matching algorithm. Then, they developed multivariate regression models with dummy variables, and analyzed the effects of elevated CO₂ on volume growth and compared the difference of effects between planted and natural stands. Focusing on the modeling and dummy variables used by the authors, it seems to be sound. The data were obtained from the USFS-FAI database, and much enough for modeling. The two groups were classified appropriately, and the best matching algorithm was used for balancing the plots. The results were presented in sufficient detail, and analyzed reasonably. However, the conclusion “the 75-ppm increase in CO₂ between 1970 and 2015 engendered a 29.6% to 35.7% increase in aboveground biomass in naturally regenerated stands and a 33.4% to 36.6% increase in planted stands” really surprised me.

We also were surprised by this result, but note that it is the effect for stands whose ages were truncated. In order to conduct a clean comparison between the naturally regenerated and planted stands, we limited the observations to those aged 1 to 50 (because there are so few older planted stands). When we compare the CO₂ effects on natural stands for stands aged 1 to 100, the parameter on cumulative life CO₂, which measures the average effect of elevated CO₂ across all the matched stands from 0 to 100, is smaller (Table S39: Lines 856-857). This is expected because a 75 ppm change in CO₂ level from 1970 to 2015 will have a larger proportional effect on younger stands.

On my opinion, the model structure might need to be taken into consideration again.

This paper aims to identify the impact of elevated CO₂ on volume growth, then a base growth function is necessary. On line 389, the authors stated to use a set of logistic growth functions, but the three models (1), (2) and (3) with dummy variables and the model (4) without dummy variables are not really growth functions, they are multivariate regression models. For a growth model, the key variable is age, other variables are less important, which can be used as dummy variable or random variable affecting the parameters of growth function. For a multivariate regression model, all variables are equally important, and affect independently the target variable such as volume. This is the difference between two kinds of models.

Thank you bringing this important issue to our attention. We were imprecise in our terminology and referred to the function we estimated as a growth function in a number of places, when in fact it is as the reviewer notes just a relationship between aboveground growing-stock volume and variables that can explain volume, including age, climate, soils, etc. We have corrected this (Lines 93, 129, and 275) and have removed references to estimating a growth function, which obviously could imply estimating a function of the change in volume from one period to the next, as the equations you suggest would do.

We wish to note that there are strong ties between the exponential volume approach and the growth function approach. The standard exponential volume function is written as:

$$V(t) = \exp(a-b/t)$$

Which can then be transformed into:

$$\ln(V(t)) = a-b/t$$

Which is the generalized version of the equation that is used in our study. If we were to then take the derivative with respect to t , we would get the following:

$$d\ln(V(t))/dt = b/t^{-2}$$

which would assess the rate of change with time or in other words the growth rate. Thus, the exponential volume and logistic growth function are drawn from the same mathematical foundation.

In the early stages as we drew up plans for this research project, we considered using a logistic growth function just as you have suggested. Growth functions are quite valuable and appropriate when estimating how periodic growth rates of forests have changed over time. We, however, decided not to use them as we wanted to tackle a different question, namely estimating how CO₂ exposure impacts growing-stock volume. So, we used the quasi-experimental techniques of matching to construct a dataset of matched plots from two time periods, where the difference in time periods gives us significant variation in lifetime CO₂ exposure to test for changes in CO₂ concentrations. Within this approach, the exponential volume approach, which is widely used in the natural resource and environmental economics field¹ and has the added benefit of providing the doubly robust assurances, seemed the best way to proceed. We regret being imprecise in our language in the text and thank the reviewer for pointing this out so that we could create an improved revision.

In this study, all the variables in equations S1 and S2 would affect the estimate of volume, not volume growth. Age is not the key variable, which affect the volume estimate together with other variables. To analyze the effect of elevated CO₂ on volume growth, we need to develop two average growth functions based on the data of treatment and control groups, and analyze whether the difference is significant or not. Also, we can develop one average growth function with a time-dummy variable. Taking the logistic function as an example, the average volume growth function is as follows:

$$V = a / (b + \exp(-c * \text{Age}))$$

And the general volume growth function with dummy variable is as follows:

$$V = (a_0 + a_i * X_i) / ((b_0 + b_i * X_i) + \exp(-(c_0 + c_i * X_i) * \text{Age}))$$

X_i can be defined as all variables of interest, such as elevated CO₂ and climate factors. If some of the a_i, b_i, and c_i are not significant, then delete them and refit the model again. In addition, the volume data exhibit heteroscedasticity, that is, the error variance is not constant over all observations. Thus, weighted regression or logarithmic regression should be used for parameter estimation of the growth models.

We agree with this approach to model growth, however as noted above, we have modeled volume at a given time period as a function of age and other variables that capture site qualities. This approach has a long history of use in the literature², usually to estimate the volume at given ages for a single site class. We add climate variables and other site characteristics to help control for the large heterogeneity in our samples.

For the purpose of developing a multivariate regression model, the GM matching algorithm should be efficient. But for developing a growth model, I can't make sure whether the balanced plots of two groups are sufficiently matched, because age is the most important variable.

Finally, it is suggested find more references about growth models.

Thanks for this comment. As noted above, we use the matching approach to construct our experiment,

¹ Conrad, Jon M. *Resource economics*. Cambridge university press, 2010.

² MacKinney, A. L., and L. E. Chaiken. "Volume, yield, and growth of loblolly pine in the Midatlantic Coastal Region." *Technical note (Appalachian Forest Experiment Station (Asheville, NC))*; 33 (1939).

which is to estimate the volume of forest plots that are matched between a treatment (post 2000) and control group (before 1990). By matching, we have identified plots in two different time periods with significantly different CO₂ exposures that are otherwise similar in observable covariates. We use the log of cumulative CO₂ exposure as our key independent variable in the post-matching regression of growing stock volume. Matching along with post-matching regression analysis with time fixed effects are used to guard against omitted variable bias. We argue that this provides a credible research design to test for the effect of carbon fertilization on growing-stock volume in the underlying forest plots. Our results indicate that carbon fertilization has increased the growing-stock volume on each plot an average of +1.1% for each 1% increase in cumulative lifetime CO₂ concentration (Lines 113-116).

The matching statistics provided in the supplement indicate an extremely balanced sample in observable covariates, suggesting good matches in FIA plots over time (Tables S9-S12, S28-S29, and S33-S35: Lines 660-677, 779-790, and 817-837). We have also tested and shown an alternative matching approach, as well as unmatched estimates with the full data (Tables S7-S8: Lines 645-656). Note that part of the reason for choosing a multivariate regression model instead of a growth model comes from the fact that matching followed by multivariate regression has support in the literature as a doubly robust econometric approach that has been shown to yield results that are robust to misspecification in either the matching or the regression model^{3,4,5,6}.

Special comments

Line 7, 18, 22-23, 134, 173, 190, 239, 250, 534, 709, and 712: “biomass” needs to be changed to “volume”, because the target variable in equation S1 or S2 is volume, not biomass.

All references to biomass have been changed to volume (e.g. Lines 5, 11, 13, 15, and 33).

Line 125: “between 11 and 15% (12 to 41 m³ha⁻¹)” need to be changed to “between 10.7% and 15.2% (12.0 to 40.7 m³ha⁻¹)”, keeping them consistent with the values in Table 1.

Changes made as suggested (Line 122).

Line 129: “+13% (+26 m³ha⁻¹)” need to be changed to “12.9% (25.6 m³ha⁻¹)”, keeping them consistent with the values in Table 1.

Changed as suggested (Line 131).

Table 1: All of the values in brackets need to keep 1 decimal digit, for example, (1.4, 29) need to be changed to (1.4, 29.0). That is, the zero “.0” can’t be omitted.

All values have been changed to preserve 1 decimal digit, even in the case of “.0” (Table 1: Line 138).

³ Imbens, G.W. and Wooldridge, J.M., 2009. Recent developments in the econometrics of program evaluation. *Journal of economic literature*, 47(1), pp.5-86.

⁴ Stuart, E.S. Matching methods for causal inference: A review and a look forward. *Statistical Science* 25, 1 (2010).

⁵ Ho, D.E., Imai, K., King, G., and Stuart, E.S. Matching as nonparametric preprocessing for reducing model dependence in parametric causal inference. *Political Analysis* 15, 199-236 (2007).

⁶ Colson, K.E., Rudolph, K.E., Zimmerman, S.C., Goin, D.E., Stuart, E.A., Van Der Laan, M. and Ahern, J., 2016. Optimizing matching and analysis combinations for estimating causal effects. *Scientific Reports*, 6(1), pp.1-11.

Line 186: “24%, 29% and 24%” need to be changed to “24.0%, 29.0% and 24.3%”, keeping them consistent with the values in Table 2.

Formatting changed to be consistent with values presented in table (Lines 165-166).

Line 188: “27%, 30% and 33%” need to be changed to “27.1%, 29.8% and 32.7%”, keeping them consistent with the values in Table 2.

Formatting changed to be consistent with values presented in table (Lines 167-168).

Line 208: What does DGVM mean? The detailed expression should be given.

The full name, dynamic global vegetation model, has been added to the text (Lines 196-197).

Line 224: We can't find 14.4% in all tables, which need to be provided in the result section.

This has been updated and direction to Table 1 has been included to aid in finding the source (Line 242).

Line 240-242: It should be pointed out it is only for 75 years old forests, and “13% (+/-2%;p<0.01)” need to be changed to “12.9% (between 11.3% and 14.5% for p<0.01)”, keeping them consistent with the values in Table 1.

Made suggested changes (Lines 206-207).

Line 362, 718-719: We can't find references numbered as 40 and 41. I guess the references 42 and 43 to be 40 and 41, respectively.

We have gone back through the entire manuscript and double-checked that all numbers cited have a corresponding entry in one of the two reference sections.

Line 454-456: Six decrease and three increase, not five and four.

This mistake has been corrected (Lines 321-322).

Line 562-628: In the tables S9-S18, the estimates of parameters should keep the same significant digits, not same decimal digits. For example, the range of significant digits for parameter estimates of model (1) in table S9 is from 1 (0.000004 for mean winter precipitation) to 9 (190.999293 for constant), this is not the scientific expression. The best way is to use scientific notation. For example, the five parameter estimates in the most front are expressed as 1.103, 5.219, 1.836, -3.212×10^{-1} , 1.620×10^{-2} , respectively. In general, 4 or 5 significant digits are enough. Same as in tables S21 and S22.

All tables have been updated in the manner suggested (e.g. Tables S7-S8: Lines 645-656).

Line 668: Figure S2 should be Figure S1.

Figure was removed from supplement.

Reviewers' Comments:

Reviewer #1:

Remarks to the Author:

The authors have sufficiently addressed my previous critiques, as well as those by the other reviewer. This is the third time I have read this article. I have no further comments or concerns.

Reviewer #2:

Remarks to the Author:

The authors have properly responded to most of my questions. I still have a nagging feeling about the robustness of the results but I can't tie it to any objective metrics. Thus, I have no major concerns at the moment but also want to emphasize that I don't feel perfectly capable of evaluating the study in all its aspects.

Given that the results might trigger interest, I would urge the authors to make the entire dataset as used in the study publicly available in a machine readable format with example code. This is the only and best way towards transparency and allowing the wider research community to a) check results and b) test robustness against other biasing variables that the authors haven't yet thought about.

Reviewer #4:

Remarks to the Author:

I have read the authors' responses to four reviewers' comments. I also think the statistical matching technique is novel, creative and informative, and the authors really improved the manuscript and provided much enough information in the supplement. However, I still think it need more study before giving the conclusion. Viewing from the modeling approach, volume model and volume growth model (or growth process model) are quite different models. In this paper, the authors developed multivariate volume models for natural and planted forests. If the R^2 was high enough, the contribution of CO₂ to the volume estimate would be reliable, but just as the Reviewer #2 said, the R^2 seemed pretty poor. If we can develop volume growth models and identify the difference of volume for same age stands on two conditions at high and low CO₂ levels, then we could analyze the impact of CO₂ to volume estimate. Thus, I still suggest the authors try to develop volume growth model using Logistic or Richards functions based on the matched samples. Taking the Logistic function as an example, the simple model is:

$$V = (a_0 + a_1 * X) / ((b_0 + b_1 * X) + \exp(-(c_0 + c_1 * X) * \text{Age})) \quad (1)$$

Where X is the dummy variable indicating treatment samples (X=0 at low CO₂ level) or control samples (X=1 at high CO₂ level).

I think the fitting result of model (1) would be helpful. If it can support the conclusion, the authors can add them in the supplement.

REVIEWER COMMENTS

Reviewer #1 (Remarks to the Author):

The authors have sufficiently addressed my previous critiques, as well as those by the other reviewer. This is the third time I have read this article. I have no further comments or concerns.

We thank the reviewer for the time and effort spent reviewing our manuscript.

Reviewer #2 (Remarks to the Author):

The authors have properly responded to most of my questions. I still have a nagging feeling about the robustness of the results but I can't tie it to any objective metrics. Thus, I have no major concerns at the moment but also want to emphasize that I don't feel perfectly capable of evaluating the study in all its aspects.

Given that the results might trigger interest, I would urge the authors to make the entire dataset as used in the study publicly available in a machine readable format with example code. This is the only and best way towards transparency and allowing the wider research community to a) check results and b) test robustness against other biasing variables that the authors haven't yet thought about.

This is a great point. We are readying the data and code in order to share them with the research community. We thank the reviewer for the help in improving our manuscript.

Reviewer #4 (Remarks to the Author):

I have read the authors' responses to four reviewers' comments. I also think the statistical matching technique is novel, creative and informative, and the authors really improved the manuscript and provided much enough information in the supplement. However, I still think it need more study before giving the conclusion. Viewing from the modeling approach, volume model and volume growth model (or growth process model) are quite different models. In this paper, the authors developed multivariate volume models for natural and planted forests. If the R^2 was high enough, the contribution of CO₂ to the volume estimate would be reliable, but just as the Reviewer #2 said, the R^2 seemed pretty poor. If we can develop volume growth models and identify the difference of volume for same age stands on two conditions at high and low CO₂ levels, then we could analyze the impact of CO₂ to volume estimate. Thus, I still suggest the authors try to develop volume growth model using Logistic or Richards functions based on the matched samples. Taking the Logistic function as an example, the simple model is:

$$V = (a_0 + a_1 * X) / ((b_0 + b_1 * X) + \exp(-(c_0 + c_1 * X) * \text{Age})) \quad (1)$$

Where X is the dummy variable indicating treatment samples (X=0 at low CO₂ level) or control samples (X=1 at high CO₂ level).

I think the fitting result of model (1) would be helpful. If it can support the conclusion, the authors can add them in the supplement.

We appreciate the reviewer's concerns and have tried to address them directly in this resubmission.

In response to your statement "To analyze the effect of elevated CO₂ on volume growth, we need to develop two average growth functions based on the data of treatment and control groups, and analyze whether the difference is significant or not... Taking the logistic function as an example, the average volume growth function is as follows: $V = a / (b + \exp(-c * \text{Age}))$ ", first, thank you for the helpful suggestion. We followed this approach in Tables S42 and S43 and results show a significant increase in volume for treatment observations relative to control ones.

Then, as you also suggested to "develop one average growth function with a time-dummy variable... And the general volume growth function with dummy variable is as follows:

$V = (a_0 + a_i * X_i) / ((b_0 + b_i * X_i) + \exp(-(c_0 + c_i * X_i) * \text{Age}))$ ", we followed this approach in Tables S44 and S45 using a logarithmic transformation of your equation (to enable better comparison with our main text results that were also logarithmically transformed) and results again showed an increase due to elevated CO₂ that was similar in sign and magnitude to those presented in the main text.

Finally, following your recommendation to take it that "X_i can be defined as all variables of interest, such as elevated CO₂ and climate factors. If some of the a_i, b_i, and c_i are not significant, then delete them and refit the

model again”, we applied this approach in Tables S46 and S47 and found once again that results appear to be robust across these choices of methodology.

We maintain the original results as the primary approach [matching in conjunction with regressions based on the log-linear volume function of the form $\text{Volume} = \exp(a + b*(1/\text{age}))$ or $\ln(\text{Volume}) = a + b*(1/\text{age})$] for the main text because it represents a doubly robust estimation approach that is robust to misspecification in either the matching or the regression model. However, we also note in the text and in the supplement that the post-matching results are similar for the non-linear volume function of the form $V = (a_0 + a_i * X_i) / ((b_0 + b_i * X_i) + \exp(-(c_0 + c_i * X_i) * \text{Age}))$. This point is made in the manuscript at the bottom of page 24.

Reviewers' Comments:

Reviewer #4:

Remarks to the Author:

The authors have sufficiently addressed my previous concerns. The section "Methods" should be moved to the front of section "Results". In addition, I think the results in Table S44 and Table S45 might be more reliable and reasonable. I have no further comments.